# INTRIGUING CLASS-WISE PROPERTIES OF ADVERSARIAL TRAINING

## ABSTRACT

Adversarial training is one of the most effective approaches to improve model robustness against adversarial examples. However, previous works mainly focus on the overall robustness of the model, and the in-depth analysis on the role of each class involved in adversarial training is still missing. In this paper, we provide the first detailed class-wise diagnosis of adversarial training on six widely used datasets, *i.e.*, MNIST, CIFAR-10, CIFAR-100, SVHN, STL-10 and ImageNet. Surprisingly, we find that there are *remarkable robustness discrepancies among classes*, demonstrating the following intriguing properties: 1) Many examples from a certain class could only be maliciously attacked to some specific semantic-similar classes, and these examples will not exist adversarial counterparts in bounded $\epsilon$-ball if we re-train the model without those specific classes; 2) The robustness of each class is positively correlated with its norm of classifier weight in deep neural networks; 3) Stronger attacks are usually more powerful for vulnerable classes. Finally, we propose an attack to better understand the defense mechanism of some state-of-the-art models from the class-wise perspective. We believe these findings can contribute to a more comprehensive understanding of adversarial training as well as further improvement of adversarial robustness.

## 1 INTRODUCTION

The existence of adversarial examples (Szegedy et al., 2014) reveals the vulnerability of deep neural networks, which greatly hinders the practical deployment of deep learning models. Adversarial training (Madry et al., 2018) has been demonstrated to be one of the most successful defense methods by Athalye et al. (2018). Some researchers (Zhang et al., 2019; Wang et al., 2019b; Carmon et al., 2019; Song et al., 2019) have further improved adversarial training through various techniques. Although these efforts have promoted the progress of adversarial training, the performance of robust models is far from satisfactory. Thus we are eager for some new perspectives to break the current dilemma.

We notice that focusing on the differences among classes has achieved great success in the research of noisy label (Wang et al., 2019a) and long-tailed data (Kang et al., 2019), while researchers in adversarial community mainly concentrate on the overall robustness. A question is then raised:

*How is the performance of each class in the adversarially robust model?*

To explore this question, we conduct extensive experiments on six commonly used datasets in adversarial training, *i.e.*, MNIST (LeCun et al., 1998), CIFAR-10 & CIFAR-100 (Krizhevsky et al., 2009), SVHN (Netzer et al., 2011), STL-10 (Coates et al., 2011) and ImageNet (Deng et al., 2009), and the pipeline of adversarial training and evaluation follows Madry et al. (2018) and Wong et al. (2019). Figure 1 plots the robustness of each class at different epochs in the test set, where the shaded area in each sub-figure represents the robustness gap between different classes across epochs. Considering the large number of classes in CIFAR-100 and ImageNet, we randomly sample 12 classes for a better indication, and the number of classes in each robustness interval is shown in Appendix A.

From Figure 1, we surprisingly find that there are recognizable robustness gaps between different classes for all datasets. Specifically, for SVHN, CIFAR-10, STL-10 and CIFAR-100, the class-wise robustness gaps are obvious and the largest gaps can reach at 40%-50% (Figure 1(b)-1(e)). For

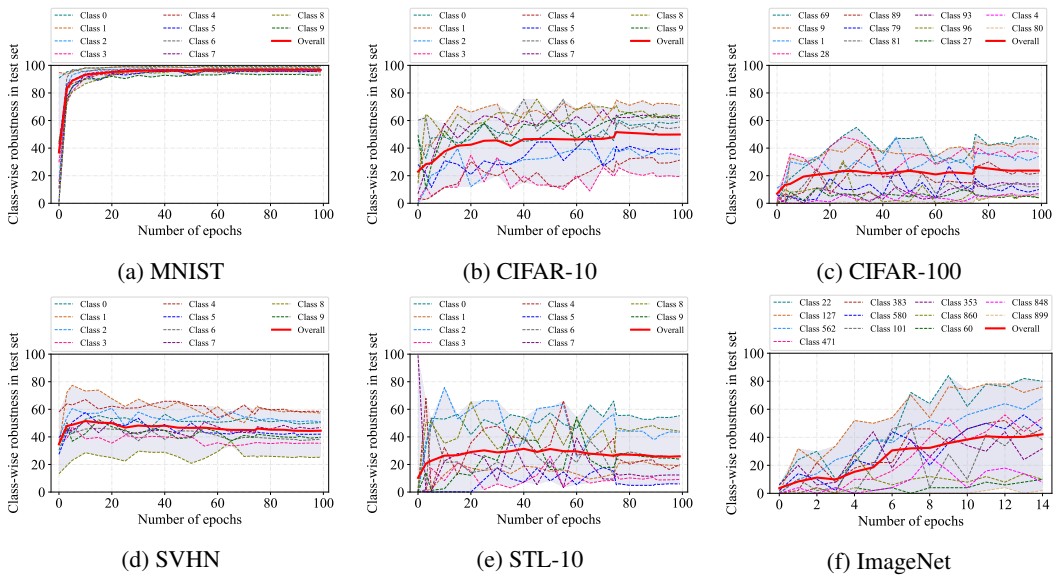

Figure 1: Class-wise robustness at different epochs in test set

ImageNet, since the model uses the three-stage training method (Wong et al., 2019), its class-wise robustness gap increases with the training epoch, and finally up to 80% (Figure 1(f)). Even for the simplest dataset MNIST, on which model has achieved more than 95% overall robustness, the largest class-wise robustness gap still has 6% (Figure 1(a)). Motivated by the above discovery, we naturally raise the following three questions to better investigate the class-wise properties in the robust model:

1) Is there any relations among these different classes as they perform differently?
2) Are there any factors related to the above phenomenon?
3) Is the class-wise performance related to the strength of the attack?

We conduct extensive analysis on the obtained robust models and gain the following insights:

- Many examples from a certain class could only be maliciously flipped to some specific classes. As long as we remove those specific classes and re-train the model, these examples will not exist adversarial counterparts in bounded $\epsilon$-ball.

- The robustness of each class is near monotonically related to its norm of classifier weight in deep neural networks.

- In both white-box and black-box settings (Dong et al., 2020), stronger attacks are usually more effective for vulnerable classes (*i.e.*, their robustness is lower than overall robustness).

Furthermore, we propose a simple but effective attack called Temperature-PGD attack. It can give us a deeper understanding of how variants of Madry's model work, especially for the robust model with obvious improvement in vulnerable classes(Wang et al., 2019b; Pang et al., 2020). Thus, our work draws the attention of future researchers to watch out the robustness discrepancies among classes.

## 2 RELATED WORK

**Adversarial training.** Adversarial training (Madry et al., 2018) is often formulated as a min-max optimization problem. The inner maximization applies the Projected Gradient Descent (PGD) attack to craft adversarial examples, and the outer minimization uses these examples as augmented data to train the model. Subsequent works are then proposed to further improve adversarial training, including introducing regularization term (Zhang et al., 2019; Wang et al., 2019b), adding unlabeled data (Carmon et al., 2019; Uesato et al., 2019; Zhai et al., 2019), and data augmentation (Song et al.,

2019). Since adversarial training is more time-consuming than standard training, several methods (Shafahi et al., 2019; Wong et al., 2019) are proposed to accelerate the adversarial training process.

**Exploring the properties in adversarial training.** A lot of researchers try to understand adversarial training from different perspectives. Schmidt et al. (2018) find more data can improve adversarial training. Tsipras et al. (2019) demonstrate that adversarial robustness may be inherently at odds with natural accuracy. Zhang & Zhu (2019) visualize the features of the robust model. Xie & Yuille (2019) explore the scalability. The work of Ortiz-Jimenez et al. (2020) is most relevant to ours. The difference is they focus on the distance from each example to the decision boundary, while we provide some new insights on the role of different classes in adversarial training.

## 3 EXPLORING THE PROPERTIES AMONG DIFFERENT CLASSES ON ADVERSARIAL TRAINING

Inspired by the phenomenon in Figure 1, we further conduct the class-wise analysis to explore the properties among different classes for a better understanding on adversarial training.

**Datasets.** We use six benchmark datasets in adversarial training to obtain the corresponding robust model, *i.e.*, MNIST (LeCun et al., 1998), CIFAR-10 & CIFAR-100 (Krizhevsky et al., 2009), SVHN (Netzer et al., 2011), STL-10 (Coates et al., 2011) and ImageNet (Deng et al., 2009). Table 1 highlights that the classes of CIFAR-10 and STL-10 can be grouped into two superclasses: *Transportation* and *Animals*. Similarly, CIFAR-100 also contains 20 superclasses with each has 5 subclasses. See Appendix B for more details of all datasets.

Table 1: Superclasses in CIFAR-10 and STL-10.

| Dataset | Transportation | | | | Animals | | | | | |
|---------|---------|---------|---------|---------|---------|---------|---------|---------|---------|---------|
| CIFAR-10 | Airplane(0)[1] | Automobile(1) | Ship(8) | Truck(9) | Bird(2) | Cat(3) | Deer(4) | Dog(5) | Frog(6) | Horse(7) |
| STL-10 | Airplane(0) | Car(2) | Ship(8) | Truck(9) | Bird(1) | Cat(3) | Deer(4) | Dog(5) | Horse(6) | Monkey(7) |

[1] The number in brackets represents the numeric label of the class in the dataset.

For ImageNet dataset, the pipeline of adversarial training follows Wong et al. (2019), while the training methods of other datasets follow Madry et al. (2018). The detailed experimental settings are:

**MNIST setup.** Following Zhang et al. (2019), we also use a four-layers CNN as the backbone. In the training phase, we adopt the SGD optimizer (Zinkevich et al., 2010) with momentum $0.9$, weight decay $2 \times 10^{-4}$ and an initial learning rate of $0.01$, which is divided by 10 at the $55^{\text{th}}$, $75^{\text{th}}$ and $90^{\text{th}}$ epoch (100 epochs in total). Both the training and testing attacker are 40-step PGD ($\text{PGD}^{40}$) with random start, maximum perturbation $\epsilon = 0.3$ and step size $\alpha = 0.01$.

**CIFAR-10 & CIFAR-100 setup.** Like Wang et al. (2019b) and Zhang et al. (2019), we also use ResNet-18 (He et al., 2016) as the backbone. In the training phase, we use the SGD optimizer with momentum $0.9$, weight decay $2 \times 10^{-4}$ and an initial learning rate of $0.1$, which is divided by 10 at the $75^{\text{th}}$ and $90^{\text{th}}$ epoch (100 epochs in total). The training and testing attackers are $\text{PGD}^{10}/\text{PGD}^{20}$ with random start, maximum perturbation $\epsilon = 0.031$ and step size $\alpha = 0.007$.

**SVHN & STL-10 setup.** All settings are the same to CIFAR-10 & CIFAR-100, except that the initial learning rate is $0.01$.

**ImageNet setup.** Following Shafahi et al. (2019) and Wong et al. (2019), we also use ResNet-50 (He et al., 2016) as the backbone. Specifically, in the training phase, we use the SGD optimizer with momentum $0.9$ and weight decay $2 \times 10^{-4}$. A three-stage learning rate schedule is used as the same with Wong et al. (2019). The training attacker is FGSM (Goodfellow et al., 2015) with random start, maximum perturbation $\epsilon = 0.007$, and the testing attacker is $\text{PGD}^{50}$ with random start, maximum perturbation $\epsilon = 0.007$ and step size $\alpha = 0.003$.

For ImageNet dataset, a $14^{\text{th}}$ epoch model is used to evaluate robustness as it did in Wong et al. (2019). For other datasets, a $75^{\text{th}}$ epoch model is used like it did in Madry et al. (2018). These settings are fixed for all experiments unless otherwise stated. This paper mainly focuses on the adversarial robustness of the model, but the comparisons of the class-wise performance between the robust model and standard model are highlighted in Appendix C for saving space.

### 3.1 The relations among different classes

We first systematically investigate the relation of different classes under robust models. Figure 2 shows the confusion matrices of robustness between classes on all the six datasets. The X-axis and Y-axis represent the predicted classes and the ground truth classes, respectively. The grids on the diagonal line represent the robustness of each class, while the grids on the off-diagonal line represent the non-robustness on one class (Y-axis) to be misclassified to another class (X-axis).

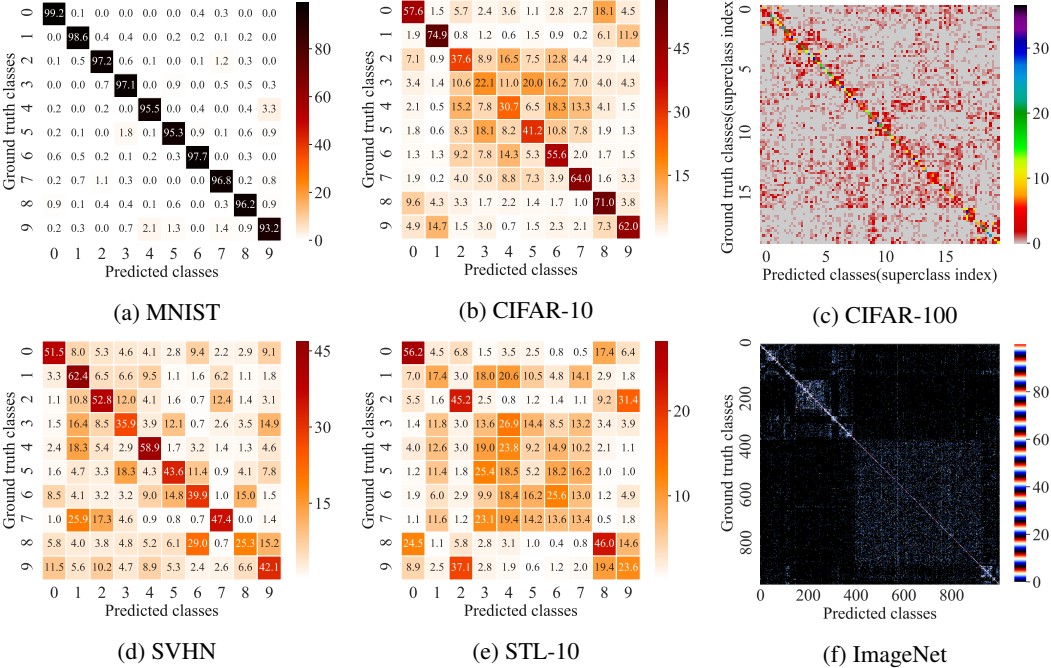

Figure 2: Confusion matrix of robustness in test set

**Results Analysis.** All the confusion matrices in Figure 2 roughly demonstrate one kind of symmetry, indicating that similar classes could easily be maliciously flipped to each other. Specifically, for SVHN, digits with similar shapes are more likely to be flipped to each other, *e.g.*, the number 6 and number 8 are similar in shape and the non-robustness between them (number 6 is misclassified to be number 8 or vice versa) is very high as shown in Figure 2(d). For CIFAR-10 and STL-10, Figures 2(b) and 2(e) clearly show that the classes belonging to the same superclass[1] have high probability to be craftly misclassified to each other, for example, both class 3 (cat) and class 5 (dog) in CIFAR-10 belong to the superclass *Animals*, the non-robustness between them is very high in Figure 2(b). In contrast, there are few flips between superclasses, since the classes belonging to different superclasses are dissimilar with few mutual influence. For example, in STL-10, the class 5 (dog) belongs to superclass *Animals*, while class 9 (truck) belongs to *Transportation*, and their non-robustness is almost 0 as shown in figure 2(e). A t-SNE (Maaten & Hinton, 2008) visualization of CIFAR-10 is reported in Appendix D shows for further supporting our above analysis and conclusions on CIFAR-10. For CIFAR-100 and ImageNet, we can also observe symmetry properties of confusion matrix in Figure 2(c) and Figure 2(f), indicating that some similar classes cloud easily be misclassified to each other. Overall, Figure 2 demonstrates that the classes with similar semantic would be easier misclassified (with higher non-robustness) to each other than those with different semantic (*e.g.*, the classes belong to different superclasses) in adversarial training.

**Removing the confound class.** Inspired by the relations among different classes as shown in Figure 2, in this subsection, we further investigate the class-wise properties of adversarial examples by removing the confound class. Specifically, for the example $x$ from class $i$ is attacked to the confound

---

[1]As we introduced in Table 1, class 0,1,8,9 belong to superclass *Transportation* and class 2,3,4,5,6,7 belong to superclass *Animals* in CIFAR-10. Similarly, class 0,2,8,9 belong to *Transportation* and class 1,3,4,5,6,7 belong to *Animals* in STL-10.

class $j$, we are curious if we remove confound class $j$ (*i.e.,* remove all examples of ground truth class $j$ in the training set) and re-train the model, will example $x$ become a robust example WITHOUT being maliciously flipped to a new confound class?

**Definition 1.** *(**Confound Class**) The output class of the model to adversarial example $x'$ is defined as the confound class of this example $x'$. (This class must be different from ground truth class)*

**Definition 2.** *(**Robust Example**) An example is defined as a robust example if it does not exist adversarial counterpart in bounded $\epsilon$-ball, saying it would be correctly classified by the model.*

**Definition 3.** *(**Homing Property**) Given an adversarial example $x'$ from class $i$ which is misclassified as the confound class $j$ by a model, this example satisfies homing property if it becomes a robust example after we re-train the model via removing confound class $j$.*

To explore the above question, we conduct extensive experiments on the popular dataset CIFAR-10 as the case study. The results are reported in Figure 3. Figure 3(a) and Figure 2(b) are similar, and the difference is that the values in Figure 3(a) represent the number of examples instead of percentage, and the diagonal elements (the number of examples correctly classified) are hidden for better visualization and comparison. Thus this figure is called as the Misclassified confusion matrix. To check the *homing property*, we alternatively remove each confound class to re-train the model and plot the results in Figure 3(b), where the element in the $i^{\text{th}}$ row and $j^{\text{th}}$ column (indexed by the classes starting from 0) indicates how many adversarial examples with ground truth class $i$ and confound class $j$ that satisfy *homing property* (*i.e.,* these examples will become robust examples after removing the confound class $j$), so this figure is defined as the Homing confusion matrix.

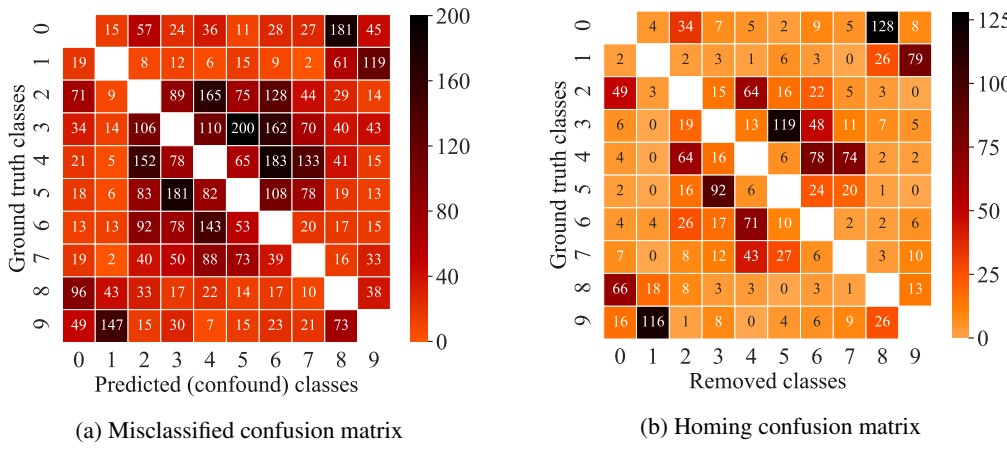

(a) Misclassified confusion matrix

(b) Homing confusion matrix

Figure 3: Misclassified and Homing Confusion matrix of CIFAR-10

**Results Analysis.** Figure 3 clearly shows *homing property* is widely observed in many misclassified examples. For example, we can focus on the $9^{\text{th}}$ row and the $1^{\text{st}}$ column of Figure 3(a) and 3(b). 147 in Figure 3(a) means that 147 examples of class 9 are misclassified as class 1, and 116 in Figure 3(b) means that if we remove class 1 and re-train the model, 116 of 147 examples will *home* to the correct class 9 (*i.e.,* become robust examples). This suggests that improving the robustness of class 9 only needs to carefully handle the relation with class 1 and has nothing to do with other classes. Interestingly, these group-based relations are commonly observed in CIFAR-10, *e.g.*, class 0 (airplane)-class 8 (ship) and class 3 (cat)-class 5 (dog).

**Conclusion and Suggestion.** Classes can be divided into several groups, and intra-group classes are easily affected by each other. More importantly, many examples from a certain class are only attacked to some specific classes (*e.g.,* most misclassified examples of class 9 are attacked to class 1), which will become robust examples if we re-train the model by removing those specific classes. This discovery inspires us that the robustness of some classes can be handled in groups, *e.g.*, group by superclass or one-to-one grouping like class 1-class 9 in figure 3(b).

## 3.2 The robustness of classes v.s. the weight of classifier

In this section, we attempt to explore whether some factors in the model are related to the unbalanced robustness among different classes in adversarial training. In fact, unbalanced performance is widely studied in the field of long-tailed data research (Wang et al., 2017), that is, the number of training examples in each class varies greatly, and the class with a small number of examples can usually achieve lower accuracy since it cannot be sufficiently trained. Recently, Kang et al. (2019) find there is a strong correlation between model parameters and the cardinality of classes, and they propose the state-of-the-art algorithm in long-tailed data research according to this property.

Inspired by their work, we naturally curious whether the robustness of each class in adversarial training is related to the model parameters. For a better demonstration, we assume that the feature dimension of the penultimate layer is $d$ and the total number of classes is $C$. Then the parameter of the last classifier can be represented by a classifier weight $\boldsymbol{W} = \{w_i\} \in \mathbb{R}^{d \times C}$ and a classifier bias $\boldsymbol{b} = \{b_i\} \in \mathbb{R}^{1 \times C}$, where $w_i \in \mathbb{R}^d$ is the classifier weight corresponding to class $i$. Similar to Kang et al. (2019), we calculate the $l_2$-norm of the classifier weight $\|w_i\|_2$ corresponding to each class $i$ ($i \in C$). To clearly show the relation between the robustness of a class $i$ and its corresponding classifier weight $w_i$, we respectively normalize its robustness and $l_2$-norm of $w_i$, and report the results in Figure 4.

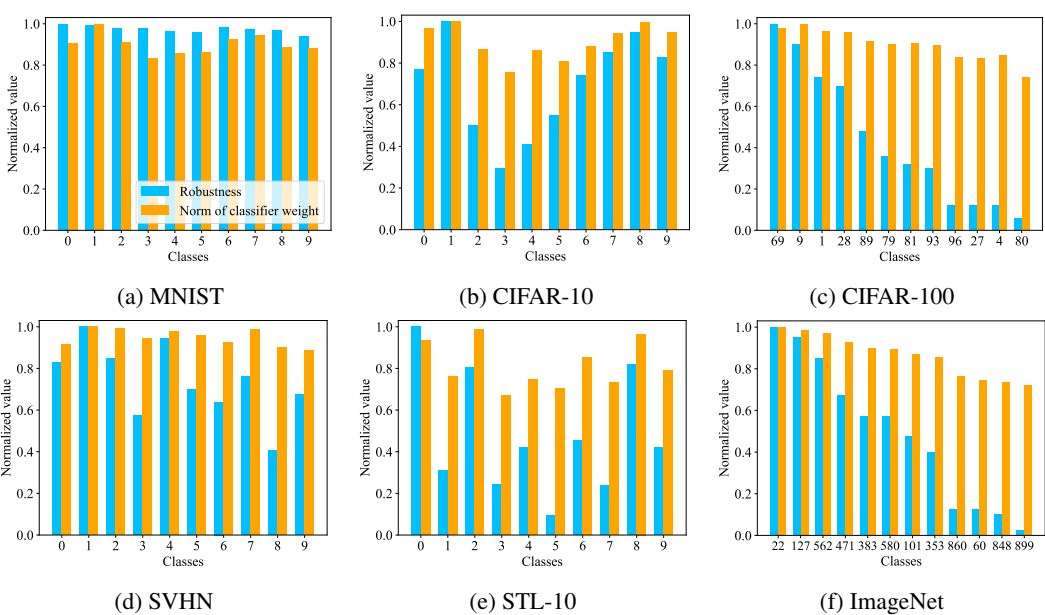

Figure 4: The correlation between the robustness of each class and its norm of classifier weight

**Results Analysis.** From Figure 4, we can find that for most classes, their robustness is positively correlated with their norm of classifier weight, *i.e.*, higher (lower) robustness corresponds to a higher (lower) norm of classifier weight. For example, the robustness of class reduced as decreasing of the norm of classifier weight across classes in CIFAR-100 as shown in 4(c). We also check this kind of correlation in standard training, but the experimental results show no significant correlation between the accuracy of each class and its corresponding norm of classifier weight in standard training. The main reason might be that these datasets in standard training are sufficient for most classes to be well trained, while adversarial training always requires abundant data for training (Schmidt et al. (2018)), hence the insufficient adversarial data cannot guarantee the classifier is well trained and lead to the above experimental observation results.

**Conclusion and Suggestion.** The robustness of each class is near monotonically related to its norm of classifier weight. Inspired by this property, we believe that balancing the norm of classifier weight of each class is a possible way to alleviate the different robustness among classes, thereby improving overall model robustness.

### 3.3 THE CLASS-WISE ROBUSTNESS UNDER DIFFERENT ATTACKS

The above analysis mainly concentrates on the performance under PGD attack. In this section, we investigate the class-wise robustness of state-of-the-art robust models against various popular attacks in the CIFAR-10 dataset.

The defense methods we chose include Madry training (Madry et al., 2018), TRADES (Zhang et al., 2019), MART (Wang et al., 2019b) and RST (Carmon et al., 2019). We re-train WideResNet-32-10 (Zagoruyko & Komodakis, 2016) following Madry et al. (2018). Other defense methods directly use the models released by the authors. White-box attacks include FGSM (Goodfellow et al., 2015), PGD (Madry et al., 2018) and $CW_\infty$ (Carlini & Wagner, 2017), and the implementation of $CW_\infty$ follows (Carmon et al., 2019). Black-box attacks include a transfer-based and a query-based attack (Dong et al., 2020). The former uses a standard trained WideResNet-32-10 as the substitute model to craft adversarial examples, and the latter uses $\mathcal{N}$ atacck (Li et al., 2019). See Appendix E for the hyperparameters of all attacks.

Table 2: Adversarial robustness (%) (under popular attacks) on CIFAR-10.

| Defenses(Attacks) | Tot. | 0 | 1 | 2 | 3 | 4 | 5 | 6 | 7 | 8 | 9 |
|---|---|---|---|---|---|---|---|---|---|---|---|
| Madry(FGSM) | 65.5 | 73.7 | 81.2[1] | 51.9 | **41.5**[2] | 54.2 | 49.4 | 73.9 | 72.5 | 78.5 | 78.5 |
| TRADES(FGSM) | 66.9 | 77.5 | 85.9 | 49.7 | **41.9** | 55.8 | 52.8 | 73.0 | 76.8 | 80.7 | 75.3 |
| MART(FGSM) | 67.4 | 73.7 | 84.9 | 54.5 | **45.7** | 50.1 | 51.6 | 76.9 | 75.2 | 83.9 | 77.7 |
| RST(FGSM) | 74.7 | 83.4 | 88.9 | 61.2 | **51.7** | 67.7 | 61.4 | 80.6 | 80.8 | 85.7 | 85.6 |
| Madry(PGD) | 52.2 | 63.8 | 71.6 | 39.1 | **25.3** | 36.7 | 38.6 | 57.4 | 59.5 | 63.1 | 66.8 |
| TRADES(PGD) | 56.4 | 67.8 | 80.6 | 37.8 | **29.4** | 40.6 | 43.9 | 59.3 | 66.9 | 71.8 | 65.6 |
| MART(PGD) | 58.3 | 64.5 | 78.0 | 45.1 | **35.4** | 37.7 | 43.5 | 65.3 | 67.5 | 76.3 | 69.5 |
| RST(PGD) | 62.9 | 74.5 | 81.8 | 46.7 | **37.2** | 46.8 | 51.0 | 66.6 | 71.9 | 74.4 | 78.0 |
| Madry($CW_\infty$) | 57.1 | 67.5 | 79.5 | 43.0 | **32.5** | 41.5 | 41.0 | 59.5 | 60.0 | 71.5 | 75.0 |
| TRADES($CW_\infty$) | 59.4 | 70.5 | 85.5 | 38.0 | **34.5** | 43.0 | 46.5 | 57.0 | 67.0 | 80.5 | 71.5 |
| MART($CW_\infty$) | 58.9 | 65.5 | 84.5 | 43.0 | **31.5** | 33.0 | 40.0 | 63.0 | 70.5 | 83.5 | 74.0 |
| RST($CW_\infty$) | 67.6 | 77.5 | 87.0 | 54.0 | **44.5** | 54.0 | 51.0 | 69.5 | 69.5 | 82.5 | 86.0 |
| Madry(Transfer-based attack) | 80.3 | 84.5 | 87.7 | 71.0 | **68.3** | 78.9 | 69.2 | 86.3 | 82.9 | 87.6 | 86.4 |
| TRADES(Transfer-based attack) | 82.0 | 87.7 | 92.3 | 70.9 | **68.0** | 78.2 | 70.0 | 87.8 | 87.6 | 90.8 | 86.8 |
| MART(Transfer-based attack) | 82.9 | 87.4 | 94.7 | 74.0 | **66.7** | 76.0 | 68.8 | 89.9 | 88.0 | 93.7 | 90.2 |
| RST(Transfer-based attack) | 88.9 | 92.5 | 95.8 | 82.4 | **75.4** | 88.8 | 79.4 | 94.1 | 91.8 | 95.2 | 93.5 |
| Madry($\mathcal{N}$ atacck) | 56.1 | 67.5 | 77.7 | 43.7 | **31.4** | 42.7 | 49.0 | 53.7 | 60.1 | 64.4 | 71.1 |
| TRADES($\mathcal{N}$ atacck) | 64.4 | 73.1 | 87.4 | 46.4 | **44.4** | 49.1 | 61.7 | 56.9 | 71.6 | 79.5 | 74.1 |
| MART($\mathcal{N}$ atacck) | 67.5 | 72.3 | 83.4 | 55.3 | **49.0** | 54.1 | 61.2 | 67.1 | 72.9 | 82.3 | 77.6 |
| RST($\mathcal{N}$ atacck) | 69.3 | 80.8 | 86.9 | 55.8 | **52.3** | 56.2 | 59.1 | 64.9 | 72.6 | 81.0 | 83.0 |

[1] The underscore indicates the most robust class.
[2] The bold indicates the most vulnerable class.

**Results Analysis.** As shown in Table 2, in all models and attacks, there are remarkable robustness gaps between different classes, which further verifies our discovery in Figure 1. Then we try to compare different attacks from a class-wise perspective. Interestingly, stronger attacks in white-box settings are usually more effective for vulnerable classes, *e.g.*, comparing FGSM and PGD, the robustness reduction of the vulnerable classes (*e.g.*, class 3) is obviously larger than that of robust classes (*e.g.*, class 1). In black-box settings, the main advantage of the query-based attack over the transfer-based attack is also concentrated in vulnerable classes. Additionally, we also notice that class 1 and class 3 are always the most robust and vulnerable class in all settings, which suggests the relative robustness of each class may have a strong correlation with the dataset itself.

**Conclusion and Suggestion.** Unbalanced robustness is commonly observed in the defenses of state-of-the-art models against popular attacks, and stronger attacks are usually more powerful for vulnerable classes, *e.g.*, class 3. We hope that future attack or defense works can report the results of class-wise robustness to better understand the proposed methods.

## 4 TEMPERATURE PGD ATTACK

Some previous work has improved significantly in the vulnerable classes, such as MART (Madry et al., 2018) in Table 2 and HE (Pang et al., 2020) in Table 5. One simple explanation is that they implicitly use instance-level (a more fine-grained level than class-level) information to improve model robustness. Specifically, MART achieves this by re-weighting misclassified examples (mainly

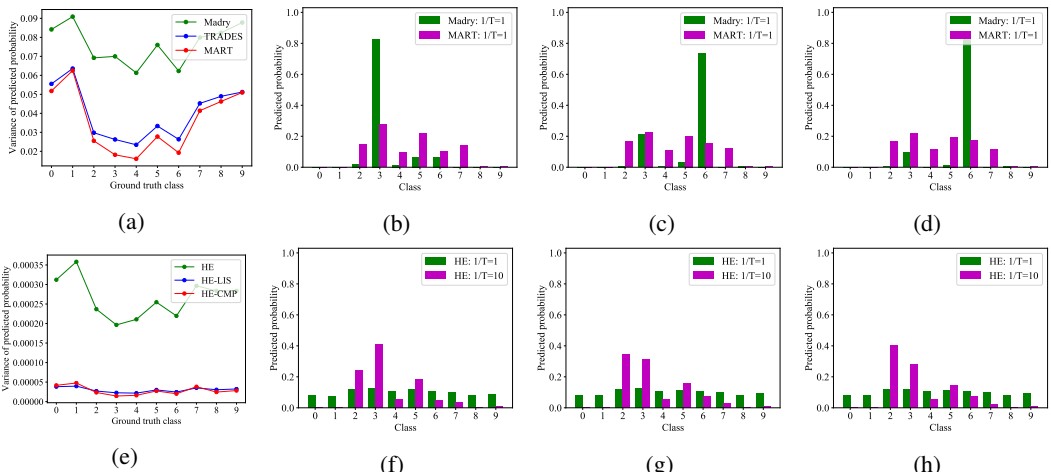

Figure 5: Analysis of class-wise predicted probability distribution: (a) Class-wise variance of predicted probability of Madry, TRADES and MART. (b)-(d) The output probability distribution (Madry: 1/T=1 and MART: 1/T=1) change of image 127 (ground truth class 3) in the process of generating adversarial examples (iteration step 1, 10 and 20). (e) Class-wise variance of predicted probability of HE, HE-LIS and HE-CMP. (f)-(h) The output probability distribution (HE: 1/T=1 and HE: 1/T=10) change of image 46 (ground truth class 3) in the process of generating adversarial examples (iteration step 1, 10 and 20).

comes from vulnerable classes) in the regular term. HE uses the re-scaled coefficient ($s$ in their paper) to make the misclassified examples provide a larger gradient for the model (more details can be found in Appendix G).

Since the misclassified examples have received higher weight during the training process, the classification boundary of each class will be more complex, and the variance of the predicted probability of example $x$ in each class $i$ ($i \in C$) will be smaller, that is, the output distribution will be more smooth as shown in Figure 5(b)-5(d) (MART). The vanilla PGD attacker may not be able to find the most effective direction in such a smooth probability distribution.

Figure 5(a) and Figure 5(e) represents the mean value of the variance of the output probability distribution for each class. Specifically, the calculation process of class 0 in the Madry's model is as follows: First, we calculate the variance of the prediction distribution of each examples with ground truth class 0 in CIFAR-10, and then use these variance to obtain the mean value. This value can well reflect the smoothness of the output distribution of each class. Combine information from Table 2, Figure 5 and Table 5, the stronger model has a lower variance than the weak model, and the vulnerable class has a lower variance than the robust class. (HE-LIS and HE-CMP are our improved methods based on HE which can boost robustness of vulnerable classes under vanilla PGD attack, see Appendix G for details). In order to find an effective direction in the extremely smooth distribution, we propose to use a temperature factor to change the smooth probability distribution, so as to create *virtual power* in the possible adversarial direction.

For a better formulation, we assume that the DNN is $f$, the input sample is $x$, the number of classes in the dataset is $C$, then the softmax probability of this sample $x$ corresponding to class $i$ ($i \in C$) is

$$\mathbb{S}(f(x))_i = \frac{e^{f(x)_i/T}}{\sum_{k=1}^{C} e^{f(x)_k/T}} \tag{1}$$

Using this improved softmax function, the adversarial perturbation crafted at $t^{\text{th}}$ step is

$$\delta^{t+1} = \prod_\epsilon (\delta^t + \alpha \cdot \text{sign}(\nabla \mathcal{L}_{CE}(\mathbb{S}(f(x + \delta^t)), y))) \tag{2}$$

Where $\prod_\epsilon$ is the projection operation, which ensures that $\delta$ is in $\epsilon$-ball. $\mathcal{L}_{CE}$ is the cross-entropy loss. $\alpha$ is the step size. $y$ is the ground truth class.

Figure 5(f)-5(h) is a good example of how Temperature-PGD works. This method allows us to arbitrarily control the absolute value of the gradient, and the idea of creating *virtual power* may

Table 3: Relative robustness (%) between Temperature-PGD[20] and vanilla PGD[20].

| Defense | 1/T | Tot. | 0 | 1 | 2 | 3 | 4 | 5 | 6 | 7 | 8 | 9 |
|---------|-----|------|---|---|---|---|---|---|---|---|---|---|
| Madry | 2 | -0.2(52.0) | -0.4[1] | +0.4[2] | -0.3 | +0.4 | -2.8 | +0.6 | -0.8 | +0.3 | -0.1 | +0.1 |
| TRADES | 5 | -1.8(54.6) | -1.0 | -0.6 | -1.1 | -3.2 | -5.0 | -0.9 | -3.3 | -0.9 | -1.0 | -0.7 |
| MART | 5 | -3.9(54.4) | -1.8 | -0.9 | -3.6 | -9.1 | -10.2 | -2.0 | -4.5 | -1.4 | -3.5 | -2.2 |
| RST | 5 | -2.0(60.9) | -1.0 | -0.6 | -2.2 | -2.1 | -4.2 | -1.5 | -5.1 | -1.1 | -1.2 | -0.8 |

[1] "-" represents the robustness reduction compared with the corresponding element of PGD attack in table 2.
[2] "+" represents the robustness improvement compared with the corresponding element of PGD attack in table 2.

also be used in targeted attack(Gowal et al., 2019), that is, manual re-scale gradients may reach the adversarial point faster in a limited number of iterations.

The data in Table 3 is the result of comparing the performance of model robustness under Temperature-PGD attack with the vanilla PGD data in Table 2. We find this method can reduce the overall robustness of the state-of-the-art models by 1.8%-3.9%, where the robustness of the vulnerable class (*i.e.,* class 4) can be reduced by 4.2%-10.2%, which is consistent with our previous findings. Since the model output of Madry's model is relatively more certain (Figure 5(b)-5(d)), the effectiveness of Temperature-PGD is not obvious. See Appendix F for more ablation studies.

In general, Temperature-PGD is a powerful tool for evaluating the defense which explicitly or implicitly use instance-wise information to improve model robustness. More importantly, it can give researchers a new perspective of how variants of Madry's model work. We speculate that the robustness improvement of many current state-of-the-art models may be due to this phenomenon. For example, label-smoothing-based defenses (Goibert & Dohmatob, 2019; Cheng et al., 2020) may not be able to defense Temperature-PGD attack, since these methods explicitly flat the distribution of predicted probabilities.

## 5 CONCLUSION

In this paper, we conduct a class-wise investigation in adversarial training based on the observation that robustness between each class has a recognizable gap, and reveal three intriguing properties among classes in the robust model: 1) Group-based relations between classes are commonly observed, and many adversarial examples satisfy the *homing property*. 2) The robustness of each class is positively correlated with its norm of classifier weight. 3) Stronger attacks are usually more effective for vulnerable classes, and we propose an attack to better understand the defense mechanism of some state-of-the-art models from the class-wise perspective. Based on these properties, we propose three promising guidelines for future improvements in model robustness: 1) Deal with different classes by groups. 2) Balance the norm of classifier weight corresponding to each class. 3) Pay more attention to classes suffering from poor robustness. We believe these findings can promote the progress of adversarial training and help to build the state-of-the-art robustness model.

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

# A  THE NUMBER OF CLASSES IN EACH ROBUSTNESS INTERVAL OF CIFAR-100 AND IMAGENET

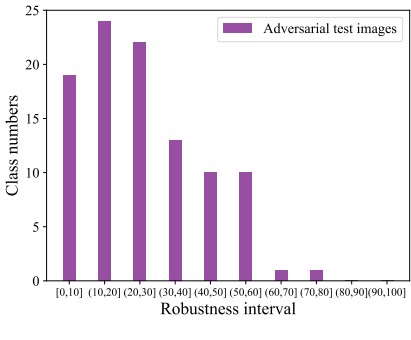
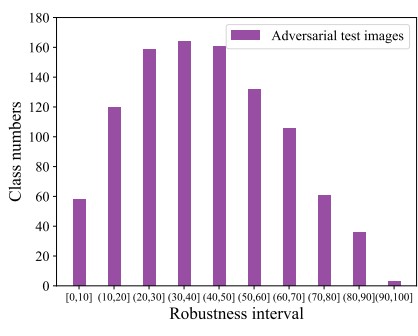

(a) CIFAR-100(75-th epoch)

(b) ImageNet(14-th epoch)

Figure 6: Number of classes per robustness interval

Due to the large number of classes in CIFAR-100 and ImageNet, we randomly sample 12 classes for analysis in the above paper. For the sake of experimental completeness, the number of classes in different robustness intervals is shown in Figure 6. Obviously, the robustness of the classes is distributed at multiple intervals, which is consistent with the results shown in Figure 1.

# B  INTRODUCTION TO THE DATASETS USED IN THE EXPERIMENT

A variety of datasets are used for research on adversarial training. Here, we introduce in detail the six datasets used in our experiment.

**MNIST.** MNIST (LeCun et al., 1998) is a handwritten digit dataset, containing numbers 0 to 9. The dataset consists of 60,000 training images and 10,000 test images, with 6,000 and 1,000 images per digit. All images are fixed size (28×28 pixels) with a value of 0 to 1, and these digits are located in the center of the image. This dataset is widely used in adversarial training(Madry et al., 2018; Zhang et al., 2019; Wang et al., 2019b; Carmon et al., 2019).

**CIFAR-10 & CIFAR-100.** CIFAR-10 & CIFAR-100 (Krizhevsky et al., 2009) are labeled subsets of the 80 million tiny images dataset (Torralba et al., 2008). CIFAR-10 consists of 50,000 training images and 10,000 test images in 10 classes, with 5,000 and 1,000 images per class. CIFAR-100 has the same total number of images as CIFAR-10, but it has 100 classes. Thus CIFAR-100 has only 500 training images and 100 test images per class. All images in these two datasets are 32×32 three-channel color images. As mentioned in Section 3, CIFAR-10 can be grouped into 2 superclasses and CIFAR-100 can be grouped into 20 superclasses. CIFAR-10 is the most popular dataset for adversarial training(Dong et al., 2020) and all proposed methods are evaluated in this dataset. CIFAR-100 is more challenging than CIFAR-10, Shafahi et al. (2019) and Song et al. (2019) evaluate their defense methods in CIFAR-100.

**SVHN.** SVHN (Netzer et al., 2011) is similar in flavor to MNIST and both of them contain 10 digits. SVHN contains 73,257 labeled digits for training, 26,032 labeled digits for testing and over 600,000 unlabeled digits images for semi-supervised or unsupervised training. In order to maintain MNIST-like style, SVHN crops the image to a size of 32×32. As a result, many of the images do contain some distractors at the sides. Due to SVHN is obtained from house numbers in the real world, its data distribution is more complicated than MNIST. Uesato et al. (2019) and Zhai et al. (2019) use this dataset for semi-supervised adversarial training.

**STL-10.** STL-10 (Coates et al., 2011) is inspired by the CIFAR-10 and sampled from images in the ImageNet (Deng et al., 2009). STL-10 also contains 10 classes and each class has only 500 training images and 800 test images. The size of all images is 96×96. As mentioned in Section 3, STL-10 can be grouped into 2 superclasses. Song et al. (2019) use this dataset to evaluate their proposed method.

**ImageNet.** ImageNet (Deng et al., 2009) is a high-resolution image dataset with 1000 classes. It contains 1,281,167 training images, 50,000 validation images and 100,000 test images. Since the test set has no labels, the validation set is often used to evaluate model robustness. The speed of training a robust model on ImageNet is intolerable, so using this dataset to evaluate model robustness usually requires some accelerated training techniques (Shafahi et al., 2019; Wong et al., 2019).

# C   COMPARE CLASS-WISE GAPS BETWEEN ADVERSARIAL TRAINING AND STANDARD TRAINING

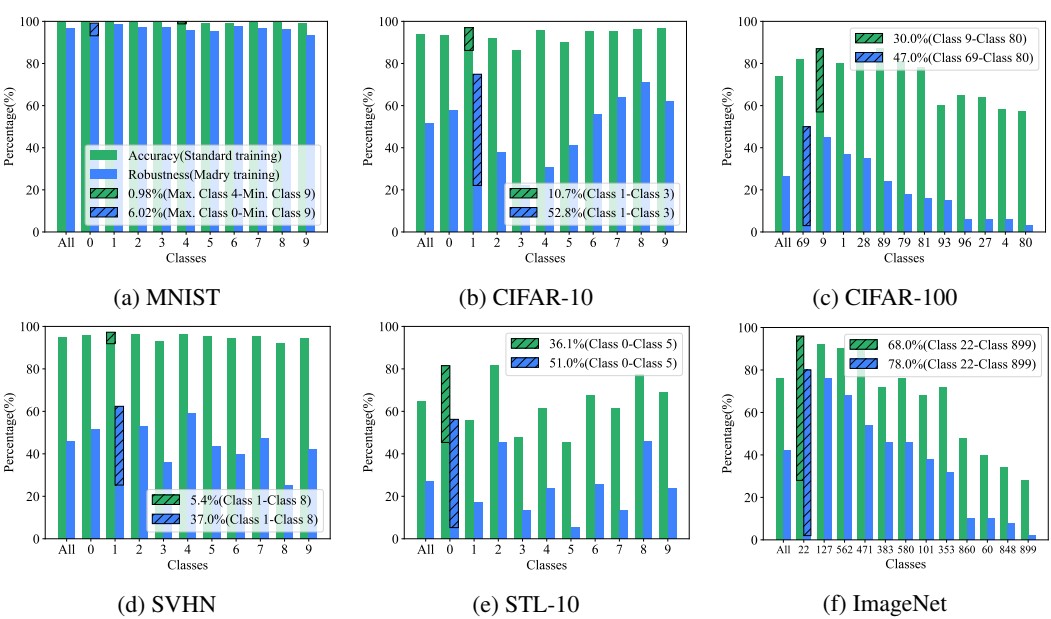

Figure 7: Class-wise accuracy in standard training and class-wise robustness in adversarial training

In the above paper, we mainly focus on the robustness gap in the model obtained by adversarial training. Since Wang et al. (2019a) also report the emergence of unbalanced accuracy in the standard model, we compare this phenomenon with that in the robust model to highlight the differences between adversarial training and standard training.

**Experimental setup.** We use the pre-trained model of Pytorch (Paszke et al., 2019) as the standard model for ImageNet. For other datasets, the experimental settings of standard models are the same as the robust models mentioned in Section 3, but adversarial examples are not added to the training set.

Figure 7 shows the class-wise accuracy in standard training and class-wise robustness in adversarial training. The slashed part in each sub-figure represents the largest gap in accuracy/robustness among different classes, and the classes in the bracket represent the highest and lowest accuracy/robustness class. Note that natural test images are evaluated for standard training, while adversarial test images are used for adversarial training.

**Results Analysis.** As illustrated in Figure 7, the relative order of accuracy/robustness of different classes is almost the same in standard training and adversarial training, but the class-wise gap is enlarged in adversarial training. For example, in CIFAR-10 and SVHN datasets, their largest class-wise accuracy gap in standard training are 10.7%(Figure 7(b)) and 5.4%(Figure 7(d)), but these indicators are enlarged to 52.8%(Figure 7(b)) and 37.0%(Figure 7(d)) in adversarial training. In more complex datasets, such as STL10, CIFAR-100 and ImageNet, although standard models also have imbalanced accuracy between classes, these gaps in adversarial training are still larger.

**Conclusion.** The performance discrepancies among classes in the robust model are larger than that of the standard model. *e.g.*, the largest gap in CIFAR-10 is enlarged from 10.7% in the standard model to 52.8% in the robust model.

# D    T-SNE VISUALIZATION OF CIFAR-10

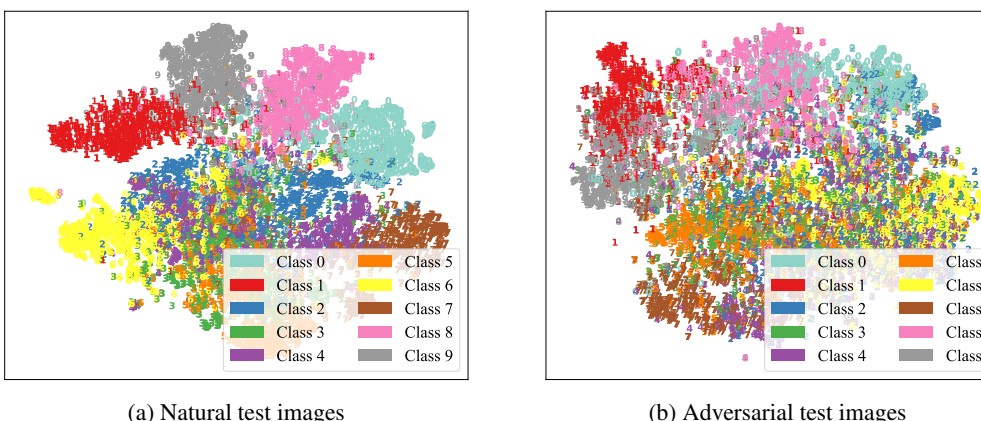

(a) Natural test images    (b) Adversarial test images

Figure 8: t-SNE visualization of CIFAR-10

In the evaluation phase, we show the t-SNE (Maaten & Hinton, 2008) visualization of CIFAR-10 to further demonstrate the property of the two superclasses *Transportation* and *Animals* in this dataset. The results of Figure 8(a) and Figure 8(b) show that the datapoints between *Transportation* (class 0,1,8,9) and *Animals* (class 2,3,4,5,6,7) can be clearly distinguished, while the datapoints between the inner classes in *Animals* are completely confused, which confirms our analysis in Section 3.1.

# E    HYPERPARAMETERS FOR POPULAR ATTACKS ON THE STATE-OF-THE-ART MODELS

**FGSM setup.** Random start, maximum perturbation $\epsilon = 0.031$.

**PGD setup.** Random start, maximum perturbation $\epsilon = 0.031$. For RST model, step size $\epsilon = 0.01$ and steps $\alpha = 40$, following Carmon et al. (2019). For other models, step size $\epsilon = 0.003$ and steps $\alpha = 20$.

**$CW_\infty$ setup.** Binary search steps $b = 5$, maximum perturbation times $n = 1000$, learning rate $lr = 0.005$, initial constant $c_0 = 0.01$, $\tau$ decrease factor $\gamma = 0.9$. Similar to Carmon et al. (2019), we randomly sample 2000 images to evaluate model robustness, and 200 images per class.

**Transfer-based attack setup.** All settings are the same to PGD for the substitute standard model.

**$\mathcal{N}$ attack setup.** Random start, maximum perturbation $\epsilon = 0.031$, population size $n_{pop} = 300$, noise standard deviation $\sigma = 0.1$ and learning rate $lr = 0.02$. Similar to Li et al. (2019), we randomly sample 2000 images to evaluate model robustness, and 200 images per class.

# F    ABLATION EXPERIMENT OF TEMPERATURE-PGD ATTACK

The data in Table 4 is similar to Table 3, except that $1/T$ is different. Combined with the results of Table 3, we can find the robustness of vulnerable classes (*e.g.*, class 4) in TRADES, MART and RST has a significant decrease in all $1/T$ settings. When $1/T$ is set to 2, the overall robustness of the model trained by Madry et al. (2018) is no exception reduced by 0.26%, with the most significant decrease by 2.8% in class 4. Furthermore, the decline of the overall robustness in Madry's model is indeed lower than that in other models, One possible explanation is that these improved robust models may obfuscate gradients Athalye et al. (2018) in vulnerable classes, and theoretical analysis is left to the future.

Table 4: Relative robustness (%) between Temperature-PGD[20] and vanilla PGD[20].

| Defense | Tot. | 0 | 1 | 2 | 3 | 4 | 5 | 6 | 7 | 8 | 9 |
|---|---|---|---|---|---|---|---|---|---|---|---|
| Madry($1/T = 2$) | -0.26 | -0.40 | +0.40 | -0.30 | +0.40 | -2.80 | +0.60 | -0.80 | +0.30 | -0.10 | +0.11 |
| TRADES($1/T = 2$) | -1.24 | -0.70 | -0.40 | -0.79 | -2.30 | -2.90 | -0.60 | -2.10 | -0.80 | -1.00 | -0.80 |
| MART($1/T = 2$) | -2.53 | -1.30 | -0.91 | -2.10 | -5.40 | -6.30 | -1.20 | -3.00 | -0.80 | -2.40 | -1.90 |
| RST($1/T = 2$) | -1.72 | -0.80 | -0.50 | -2.30 | -2.40 | -3.50 | -1.40 | -3.40 | -0.99 | -1.00 | -0.90 |
| Madry($1/T = 3$) | +0.40 | +0.10 | +1.00 | -0.10 | +0.10 | -2.60 | +1.10 | -0.20 | +2.00 | +1.50 | +1.11 |
| TRADES($1/T = 3$) | -1.68 | -0.90 | -0.50 | -1.09 | -3.20 | -4.30 | -1.10 | -2.70 | -1.11 | -1.20 | -0.70 |
| MART($1/T = 3$) | -3.28 | -1.90 | -1.00 | -3.00 | -7.40 | -8.20 | -1.60 | -3.60 | -1.21 | -2.90 | -2.00 |
| RST($1/T = 3$) | -1.96 | -1.10 | -0.70 | -2.10 | -2.00 | -4.20 | -1.30 | -5.10 | -1.09 | -1.10 | -0.90 |
| Madry($1/T = 4$) | +1.16 | +0.80 | +1.60 | +0.40 | +1.00 | -1.61 | +1.40 | +0.50 | +3.30 | +2.10 | +2.11 |
| TRADES($1/T = 4$) | -1.77 | -1.01 | -0.50 | -1.09 | -3.10 | -4.70 | -1.21 | -3.00 | -1.00 | -1.10 | -1.00 |
| MART($1/T = 4$) | -3.73 | -1.80 | -1.00 | -3.50 | -8.40 | -9.70 | -1.90 | -4.00 | -1.40 | -3.30 | -2.30 |
| RST($1/T = 4$) | -2.04 | -1.00 | -0.60 | -2.41 | -2.20 | -4.20 | -1.60 | -5.20 | -1.09 | -1.30 | -0.80 |
| Madry($1/T = 7$) | +2.97 | +2.40 | +2.10 | +1.60 | +2.90 | +0.20 | +3.70 | +4.20 | +5.30 | +4.30 | +3.01 |
| TRADES($1/T = 7$) | -1.69 | -0.70 | +0.00 | -1.19 | -3.40 | -5.10 | -0.90 | -3.40 | -0.80 | -0.90 | -0.50 |
| MART($1/T = 7$) | -3.85 | -1.80 | -0.70 | -3.70 | -8.90 | -10.90 | -2.10 | -4.60 | -1.21 | -3.20 | -1.40 |
| RST($1/T = 7$) | -1.84 | -0.60 | -0.40 | -1.90 | -2.00 | -4.30 | -1.50 | -5.00 | -0.79 | -1.00 | -0.90 |
| Madry($1/T = 10$) | +4.96 | +3.20 | +3.20 | +3.70 | +5.50 | +3.30 | +6.60 | +8.20 | +6.20 | +5.70 | +4.01 |
| TRADES($1/T = 10$) | -1.15 | +0.30 | +0.50 | -0.89 | -2.70 | -4.70 | -0.60 | -2.90 | +0.00 | -0.50 | +0.00 |
| MART($1/T = 10$) | -3.29 | -1.30 | +0.60 | -3.50 | -9.00 | -10.30 | -1.80 | -4.70 | -0.10 | -2.30 | -0.50 |
| RST($1/T = 10$) | -1.60 | -0.70 | +0.00 | -1.70 | -1.90 | -4.00 | -1.30 | -4.80 | -0.49 | -0.30 | -0.80 |

## G  IMPROVE THE DEFENSE OF PANG ET AL. (2020)

Recently, Pang et al. (2020) combine feature normalization(FN) (Ranjan et al., 2017), weight normalization(WN) (Guo & Zhang, 2017) and angular margins(AM) (Liu et al., 2016) to propose a defense method that can boost model robustness. In their paper, they believe that using FN and WN to limit embeddings on the hypersphere is the key to robustness improvement, so they call their method **hypersphere embedding(HE)**. However, we find that the factor (temperature factor in our paper) for scaling WN is the real key to improve robustness. We try to understand this from a point of view they overlooked.

Our analysis mainly focuses on FN and WN, following Pang et al. (2020). For a better formulation, we assume the input sample is $x$, the extracted feature of an example in the penultimate layer is $z \in \mathbb{R}^d$, the number of classes in the dataset is $C$, the scale factor is $s$ (corresponds to the $1/T$ in our paper) and the parameter of the last classifier is $\boldsymbol{W} = \{w_i\} \in \mathbb{R}^{d \times C}$, where $w_i \in \mathbb{R}^d$ is the classifier weight corresponding to class $i$. Then FN operation is $\widetilde{z} = z/\|z\|_2$, WN operation is $\widetilde{w_i} = w_i/\|w_i\|_2$ and the probability $P_i$ of this sample $x$ corresponding to class $i$ ($i \in C$) after the softmax function $\mathbb{S}$ is

$$P_i = \mathbb{S}(z) = \frac{e^{\widetilde{w_i}^{\mathsf{T}} \cdot \widetilde{z} \cdot s}}{\sum_{k=1}^{C} e^{\widetilde{w_k}^{\mathsf{T}} \cdot \widetilde{z} \cdot s}} = \frac{e^{\widetilde{w_i}^{\mathsf{T}} \cdot \widetilde{z}/T}}{\sum_{k=1}^{C} e^{\widetilde{w_k}^{\mathsf{T}} \cdot \widetilde{z}/T}} \tag{3}$$

The gradient of cross-entropy loss $\mathcal{L}_{CE}$ corresponding to $w_i$ is

$$\nabla_{w_i} \mathcal{L}_{CE} = \begin{cases} (P_i - 1) \cdot z & i = y \\ P_i \cdot z & i \neq y \end{cases} \tag{4}$$

Equation (3) and Equation (4) suggest we can change $P_i$ by adjusting $T$ and finally control $\nabla_{w_i} \mathcal{L}_{CE}$. *e.g.*, the maximum probability for an example is class $l$, then $P_l = 1$ if $1/T \to \infty$ and $P_l = 1/C$ if $1/T \to 0$.

For better demonstration, suppose that there is a three classification task (Figure 9). The top three figures represent equation (3) and the bottom three figures represent the corresponding equation (4). The light-colored bars in the top three figures indicate the sum of the probabilities should be equal to 1. The bars corresponding to the pink arrows in Figure 9(b) and Figure 9(e) have the same length, and Figure 9(c) and Figure 9(f) are similar.

Specifically, we consider an adversarial example, *i.e.*, the highest probability of this example is usually the incorrect class. Figures 9(a) and 9(d) represent the probability and gradient of this example after softmax. When $1/T > 1$ (Figures 9(b) and 9(e)), this example is considered by the model as a *harder* example (the probability of the adversarial class becomes larger), so that the

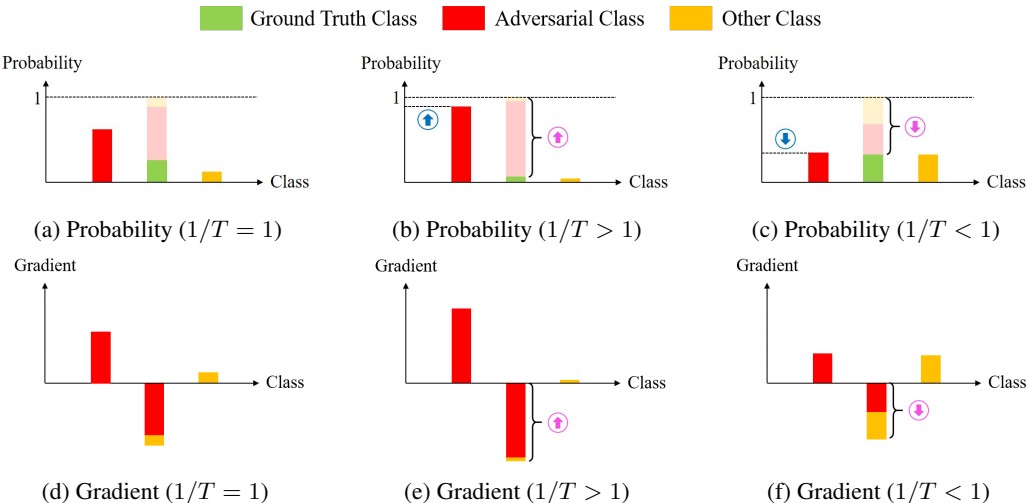

Figure 9: The relation between the probability and gradient for an adversarial example

model can update more gradients based on this example. Similarly, when $1/T < 1$ (Figures 9(c) and 9(f)), this example becomes *simpler* for the model and obtains a smaller gradient. Overall, these figures clearly show that we can adjust the gradient through the temperature factor $1/T$.

According to the above analysis and learn from the ideas of curriculum learning (Bengio et al., 2009), we believe the gradient should be gradually enlarged (by increasing $1/T$) during adversarial training, instead of using a fixed $1/T$ like Pang et al. (2020). Specifically, the following two schedules to adjust $1/T$ are proposed:

**Linear Interpolation Schedule(LIS).** We use a simple linear interpolation schedule to adjust $1/T$. Therefore, the $1/T$ of $n^{\text{th}}$ epoch is

$$\frac{1}{T_n} = \frac{1}{T_0} + \frac{n}{n_{FI}}(\frac{1}{T_{FI}} - \frac{1}{T_0}) \tag{5}$$

In our implementation, the initial temperature factor $1/T_0 = 1$ and the final temperature factor of the interpolation $1/T_{FI} = 75$, where the subscript *FI* is short for *final interpolation*. The final epoch of the interpolation is equal to the total training epoch $n_{FI} = n_{tot} = 100$. The training pipeline follows Pang et al. (2020), but we do not use angular margins. Other hyperparameters are the same to Section 3.

**Control Maximum Probability(CMP).** At each epoch, we can accurately calculate the required $1/T$ according to equation (3) to control the maximum probability. Since equation (3) is a nonlinear function, Powell's dogleg method (Powell, 1970) is used to solve this function. Therefore, the maximum probability $P^{max}$ of all examples in $n^{th}$ epoch is

$$\begin{cases} P_n^{max} = P_0^{max} + \dfrac{n}{n_{FI}}(P_{FI}^{max} - P_0^{max}) & 0 \le n < n_{FI} \\ P_n^{max} = 1 & n_{FI} \le n < n_{tot} \end{cases} \tag{6}$$

In our implementation, the initial maximum probability $P_0^{max} = 0.2$ and the final maximum probability of the interpolation $P_{FI}^{max} = 1$. The final epoch of the interpolation $n_{FI} = 90$, while the total training epoch $n_{tot} = 100$. For the last 10 epochs, the maximum probability is always controlled at 1. Other settings are the same to Linear Interpolation Schedule.

We use the above methods to train ResNet-18 on CIFAR-10 and choose vanilla PGD[20] attack to evaluate model robustness. For a fair comparison, we also re-train ResNet18 following Madry et al. (2018) and Pang et al. (2020). As shown in Table 5, our methods HE-LIS and HE-CMP have further boosted the model robustness. Especially, the improvement of vulnerable classes (*e.g.,* class 3 and class 4) is impressive. This seems to be an exciting result because the robustness gaps between classes are largely reduced.

Table 5: Adversarial robustness (accuracy (%) on vanilla PGD$^{20}$) on CIFAR-10.

| Defense | Nat. | Adv. | 0 | 1 | 2 | 3 | 4 | 5 | 6 | 7 | 8 | 9 |
|---|---|---|---|---|---|---|---|---|---|---|---|---|
| Madry | 84.0 | 51.7 | 57.6 | 74.9 | 37.6 | 22.1 | 30.7 | 41.2 | 55.6 | 64.0 | 71.0 | 62.0 |
| HE | 83.1 | 60.7 | +7.2[1] | +2.1 | +3.2 | +12.2 | +19.1 | +10.4 | +11.8 | +6.6 | +6.2 | +11.2 |
| Ours(HE-LIS) | 84.6 | 63.9 | +9.7 | +2.3 | +10.6 | +20.6 | +31.5 | +10.1 | +15.8 | +3.7 | +6.2 | +12.1 |
| Ours(HE-CMP) | 84.4 | 69.9 | +14.3 | +4.0 | +14.6 | +29.5 | +41.1 | +18.1 | +22.7 | +6.7 | +13.0 | +18.7 |

[1] "+" represents the robustness improvement compared with the corresponding element of the Madry's model.

Unfortunately, these improved methods cannot defend the Temperature-PGD attack. As shown in Table 6, the numbers in the first row indicate the different $1/T$. The improvement disappears when $1/T$ in Temperature-PGD is large enough. We have analyzed it in Section 4

Table 6: Adversarial robustness on Temperature-PGD$^{20}$ of different $1/T$

| Defense | 1 | 5 | 10 | 50 | 75 | 100 | 125 | 150 | 200 | 300 |
|---|---|---|---|---|---|---|---|---|---|---|
| Madry | 51.7 | 50.8 | 55.2 | 68.1 | 71.0 | 73.1 | 74.4 | 75.6 | 77.4 | 79.3 |
| HE | 60.7 | 57.2 | 54.1 | 50.4 | 50.7 | 51.2 | 51.7 | 52.3 | 53.2 | 55.3 |
| Ours(HE-LIS) | 63.9 | 62.7 | 60.9 | 51.9 | 50.7 | 50.3 | 50.1 | 50.1 | 50.5 | 51.7 |
| Ours(HE-CMP) | 69.9 | 67.8 | 65.1 | 54.1 | 52.2 | 51.2 | 50.7 | 50.4 | 50.3 | 50.9 |

