# OpenReview forum: "Intriguing class-wise properties of adversarial training"
_ICLR.cc/2021/Conference — Reject_

### Official Review · AnonReviewer3 · 2020-10-27
**An interesting observation, requires further investigation**

**Rating:** 4
**Confidence:** 5

**Review:**

The authors study the robustness of adversially-trained models across different classes. They find that classes tend to have largely non-uniform robust accuracy--i.e., some are less robust then others. Moreover, certain datapoints can only be misclassified as specific classes and removing these classes during training can make these datapoints robust. Next, the authors investigate how this robustness discrepancy across classes relates to the norm of the last fully-connected layer of the model, as well as to the strength of the attack used. Finally, the authors propose a new adversarial attack that they evaluate against existing models.

The observation that robust models can have widely varying robust accuracy across classes is quite interesting. At the same time, the insights gained from the current study are not particularly impactful. Specifically:
- **Robust accuracy disparity.** Before drawing conclusions, it is important to perform the same study on a standard model. Are standard class accuracies on these datasets closer to uniform? Note that this will require accounting for the variance introduced by the overall lower accuracy of robust models.
- **Certain examples can only be flipped to specific classes.** Why is this surprising? It is natural to expect that classes that are similar to each other will be more likely to be flipped on multiple examples. As a result, removing one of these classes completely from the dataset will result in the other class becoming more robust.
- **Class robustness is tied to classifier norm.** First of all, the correlation of these quantities needs to be more formally investigated. Clearly, this is not a perfect correlation so we need at least some quantitative analysis to support this conclusion. Nevertheless, it is not clear what conclusion we can draw from this finding. Is the larger norm a cause or a symptom of brittleness? Can we make certain classes more robust by reducing their weight norm? Does this allow us to understand something fundamentally new about robust models?
- **Temperature scaling attack.** It is not immediately clear what the intuition behind this attack is. From what I understand, it will cause PGD to focus more on classes with higher (original) softmax probabilities. However this is not a fundamentally new attack. Moreover, if this viewpoint is correct, its performance should be lower than that of an attack that exhaustively considers all classes and tries to produce a targeted misclassification (e.g., MultiTargeted https://arxiv.org/abs/1910.09338).

Overall, while the topic and initial exploration is interesting, the findings do not provide us with a fundamentally new understanding of robust models.

======= POST-RESPONSE UPDATE ========

I appreciate the authors' response and additional experiments. Unfortunately, my criticism still stands:
- **Comparison with standard models.** Based on Appendix C, it is still unclear if the class-wise disparity is a unique property of robust models. As we can see in Figure 7, standard models also have disparate accuracies between classes. More importantly, comparing the **absolute difference** between class accuracies is misleading, since robust models have overall lower robust accuracy. A better comparison would be to measure the **relative error** between classes. While it is hard to draw conclusions by just inspecting the graph, it seems that the discrepancy is significantly milder based on this metric.
- **Classifier norm.** The additional experiments still do not demonstrate any causal link between classifier norm and robustness. It is thus still unclear what this metric conveys.
- **Temp-PGD.** While I appreciate the effort to provide additional intuition about the attack, I still do not find the method fundamentally new when compared to other attacks optimizing combinations of logits (e.g., Carlini-Wagner, Multi-Targeted).

Overall, I still think that the original observation is intriguing, yet requires a deeper and more systematic study.

---

> ### Author Response · Authors · 2020-11-24
> **Response to AnonReviewer 3: Answering Specific Questions**
>
> **Q(1) Cons about robust accuracy disparity.**
> **A(1)** We admit that for the same robust model, their class-wise standard accuracy (the follow-up is called accuracy) and robustness accuracy (the follow-up is called robustness) trends are the same, after all, their decision boundaries are the same. However, we believe that standard training should mainly focus on accuracy (robustness is 0), and adversarial training should mainly focus on robustness. Fig.6 emphasizes that the performance discrepancies among classes in the robust model (for robustness) are larger than that of the standard model (for accuracy), and we believe this phenomenon is worthy of attention.
>
> **Q(2) Cons about flipped classes.**
> **A(2)** We usually think that there are many decision boundaries in bounded $\varepsilon$-ball, which means that a example may be flipped to many classes, but our experiment (Figure 3) find that many adversarial examples M(ground truth class is i) may only flip to a specific class j, that is, the classification boundary of examples M in $\varepsilon$-ball is unique. Once the class j is removed, these examples will become robust, that is, **it will not be flipped to a new class k.** For example, many samples of "class 9" and "class 1" in CIFAR-10 satisfy this (homing) property (see results analysis of Figure 3). We also recommend AnonReviewer3 to refer to our answer A3 for AnonReviewer1.
>
> **Q(3) Cons about classifier norm.**
> **A(3)** We have adjusted the relationship between class-wise robustness and the norm of classifier weight from causal link to related link. We will explore their causal link in future work. But we emphasize that this phenomenon itself also helps researchers better understand the robust model.  In Appendix H, we report a preliminary method based on [3], which greatly improved the robustness of vulnerable classes as shown in Table 5. Then we conduct experiments and find that the norm of classifier weight of vulnerable classes (i.e. class 3 and class 4) has significantly improved. (We also recommend AnonReviewer4 to refer to our answer A2 for AnonReviewer2.)
>
> **Madry’s model**
> **Class-wise robustness(Absolute value %):**
>
> | Class 0 | Class 1 | Class 2 | Class 3 | Class 4 | Class 5 | Class 6 | Class 7 | Class 8 | Class 9 |
> |  ----  | ----  |   ----  | ----  | ----  | ----  | ----  | ----  | ----  | ----  |
> | 57.6 | 74.9 |  37.6 | **22.1** | **30.7** | 41.2 | 55.6 | 64.0 | 71.0 | 62.0 |
>
> **Norm of classifier weight(Absolute value):**
>
> |Class 0| Class 1| Class 2| **Class 3**| **Class 4**| Class 5| Class 6| Class 7| Class 8| Class 9  |
> |  ----  | ----  |   ----  | ----  | ----  | ----  | ----  | ----  | ----  | ----  |
> |3.9399| 4.0705| 3.5359|**3.0687**| **3.5028**| 3.2904| 3.5884| 3.8438| 4.0630| 3.8535  |
>
> **Norm of classifier weight(Normalized value):**
>
> | Class 0| Class 1| Class 2| **Class 3**| **Class 4**| Class 5| Class 6| Class 7| Class 8| Class 9  |
> |  ----  | ----  |   ----  | ----  | ----  | ----  | ----  | ----  | ----  | ----  |
> | 0.96792| 1.0| 0.86867| **0.75388**| **0.86054**| 0.80835| 0.88157| 0.94429| 0.99814| 0.94669  |
>
> **HE-CMP**
> **Class-wise robustness(Absolute value %):**
>
> |Class 0| Class 1| Class 2|**Class 3**| **Class 4**| Class 5| Class 6| Class 7| Class 8| Class 9  |
> |  ----  | ----  |   ----  | ----  | ----  | ----  | ----  | ----  | ----  | ----  |
> |71.9| 78.9| 52.2| **51.6**| **71.8**| 59.3| 78.3| 70.7| 84. | 80.7  |
>
> **Norm of classifier weight(Absolute value):**
>
> |Class 0| Class 1| Class 2| **Class 3**| **Class 4**| Class 5| Class 6| Class 7| Class 8| Class 9|
> |  ----  | ----  |   ----  | ----  | ----  | ----  | ----  | ----  | ----  | ----  |
> |4.0331| 3.5757| 4.2669| **4.3090**| **4.2630**| 4.1211| 4.0952| 3.8379| 3.9430| 3.7932|
>
> **Norm of classifier weight(Normalized value):**
>
> |Class 0| Class 1| Class 2| **Class 3**| **Class 4**| Class 5| Class 6| Class 7| Class 8| Class 9|
> |  ----  | ----  |   ----  | ----  | ----  | ----  | ----  | ----  | ----  | ----  |
> |0.93598| 0.82981| 0.99022| **1.0**|**0.98932**| 0.95640| 0.95040| 0.89069| 0.91507| 0.88030|
>
>
> **Q(4) Cons about Temperature scaling attack.**
> **A(4)** Vanilla PGD is inefficient for some specific defense methods, because these methods implicitly or explicitly use instance-level (hard examples) information to smooth the prediction distribution of the model output, while Temperature-PGD can create virtual power to move the example to the nearest decision boundary. We re-introduce the motivation of Temperature-PGD in Section 4. See updated paper for details.

---

### Official Review · AnonReviewer4 · 2020-10-27
**Recommendation to Reject**

**Rating:** 4
**Confidence:** 4

**Review:**

This paper empirically studies the class-wise properties of classification models produced by existing adversarial training methods on benchmark image datasets. In particular, it demonstrates the following observations: 1) robustness varies for different seed-target class pairs; 2) for certain class, retraining the model without other semantic-similar class improves its robustness; 3) class-wise robustness relates to the norm of the model weights; 4) stronger attacks are more powerful for vulnerable classes.


Pros:

+ The paper conducts extensive experiments, including six image benchmarks.

+ Heatmaps of class-wise robustness is visually good.

+ The fine-grained analysis on class-wise properties help understand the model performance to some degree.


Cons:

- The motivation of this paper seems somewhat unclear. The paper motivates its main research question based on the observation that most existing works in adversarial community focused on overall robustness. However, it is unclear to me why focusing on overall robustness is inferior. From my perspective, it is important to clarify the following questions in the introduction: 1) If you aim to improve the overall robustness using class-wise properties, what are the potential solutions? 2) If you think overall robustness is not the best evaluation criteria for adversarial robustness, what are the reasons and alternative criteria?

- The related work section simply listed many existing works related to adversarial training. The connections between these works and your work are not well-explained. More specifically, how will the understanding of class-wise properties contribute to the improvement of existing works is unclear.

- There is a lack of coherence in Sections 3 and 4. The subsections demonstrated in Section 3 are somewhat disconnected with each other. For instance, the transition from Section 3.1 to Section 3.2 is not well-explained. In addition, Section 4 really confuses me, i.e., the proposed Temperature PGD Attack seems not related to the previous sections and does not have an advantage over the vanilla PGD attack.


Minor comments:

- It is hard for me to see a strong positive correlation from Figure 4. I would recommend the authors to compute the correlation coefficient or conduct a correlation test to support the argument.

- The paper will be much stronger if the authors can improve the performance of existing robust models by leveraging the class-wise properties discovered in Section 3.

======= POST-RESPONSE UPDATE ========

I appreciate the author's efforts for responding my questions and providing additional results and I do find the empirical observations of class-wise properties interesting. However, I still feel that the contribution of the current form of the paper is not strong enough to reach the bar of ICLR, so I remain my previous rating. Beyond exploratory analysis, the paper would be much stronger if it can go deeper with the observed class-wise properties of robust models.

---

> ### Author Response · Authors · 2020-11-24
> **Response to AnonReviewer 4: Answering Specific Questions**
>
> **Q(1) Cons about motivation.**
> **A(1)** The main purpose of this paper is to highlight the importance of class-wise robustness in adversarial community by in-depth analysis on the role of each class involved in adversarial training, such as the relationship between classes and the impact of different attacks on different classes. Such exploratory and discovery papers are common in adversarial community, such as [1] visualizes the features of robustness model, [2] explores the impact of BN on adversarial training, etc. We also report a preliminary attempt based on [3] in Appendix H and the results are shown in Table 5, which greatly improved robustness of vulnerable classes.
>
> **Q(2) Cons about related work.**
> **A(2)** In Section 2, the first part lists the related work of adversarial training. Since we did not find work similar to ours in the adversarial setting before, the second part lists the same type of papers exploring adversarial training. After other reviewers' reminders, we have added some related works.
>
> **Q(3) Cons about the coherence in Sections 3 and 4.**
> **A(3)** Section 4 introduces some defense methods that explicitly or implicitly use instance-level (a more fine-grained level than class-level) information to improve model robustness, especially the robustness of vulnerable classes can be greatly improved. Vanilla PGD cannot attack these defenses very well, while Temperature-PGD is effective. We re-introduce the motivation of Temperature-PGD in Section 4. See updated paper for details.
>
> **Q(4) Cons about Figure 4.**
> **A(4)** In Appendix H, we report a preliminary method based on [3], which greatly improved the robustness of vulnerable classes as shown in Table 5. Furthermore, we conduct experiments and find that the norm of classifier weight of vulnerable classes (i.e. class 3 and class 4) has significantly improved. (We also recommend AnonReviewer4 to refer to our answer A2 for AnonReviewer2.)
>
> **Madry’s model**
> **Class-wise robustness(Absolute value %):**
>
> | Class 0 | Class 1 | Class 2 | Class 3 | Class 4 | Class 5 | Class 6 | Class 7 | Class 8 | Class 9 |
> |  ----  | ----  |   ----  | ----  | ----  | ----  | ----  | ----  | ----  | ----  |
> | 57.6 | 74.9 |  37.6 | **22.1** | **30.7** | 41.2 | 55.6 | 64.0 | 71.0 | 62.0 |
>
> **Norm of classifier weight(Absolute value):**
>
> |Class 0| Class 1| Class 2| **Class 3**| **Class 4**| Class 5| Class 6| Class 7| Class 8| Class 9  |
> |  ----  | ----  |   ----  | ----  | ----  | ----  | ----  | ----  | ----  | ----  |
> |3.9399| 4.0705| 3.5359|**3.0687**| **3.5028**| 3.2904| 3.5884| 3.8438| 4.0630| 3.8535  |
>
> **Norm of classifier weight(Normalized value):**
>
> | Class 0| Class 1| Class 2| **Class 3**| **Class 4**| Class 5| Class 6| Class 7| Class 8| Class 9  |
> |  ----  | ----  |   ----  | ----  | ----  | ----  | ----  | ----  | ----  | ----  |
> | 0.96792| 1.0| 0.86867| **0.75388**| **0.86054**| 0.80835| 0.88157| 0.94429| 0.99814| 0.94669  |
>
> **HE-CMP**
> **Class-wise robustness(Absolute value %):**
>
> |Class 0| Class 1| Class 2|**Class 3**| **Class 4**| Class 5| Class 6| Class 7| Class 8| Class 9  |
> |  ----  | ----  |   ----  | ----  | ----  | ----  | ----  | ----  | ----  | ----  |
> |71.9| 78.9| 52.2| **51.6**| **71.8**| 59.3| 78.3| 70.7| 84. | 80.7  |
>
> **Norm of classifier weight(Absolute value):**
>
> |Class 0| Class 1| Class 2| **Class 3**| **Class 4**| Class 5| Class 6| Class 7| Class 8| Class 9|
> |  ----  | ----  |   ----  | ----  | ----  | ----  | ----  | ----  | ----  | ----  |
> |4.0331| 3.5757| 4.2669| **4.3090**| **4.2630**| 4.1211| 4.0952| 3.8379| 3.9430| 3.7932|
>
> **Norm of classifier weight(Normalized value):**
>
> |Class 0| Class 1| Class 2| **Class 3**| **Class 4**| Class 5| Class 6| Class 7| Class 8| Class 9|
> |  ----  | ----  |   ----  | ----  | ----  | ----  | ----  | ----  | ----  | ----  |
> |0.93598| 0.82981| 0.99022| **1.0**|**0.98932**| 0.95640| 0.95040| 0.89069| 0.91507| 0.88030|
>
>
> [1] Zhang T, Zhu Z. Interpreting adversarially trained convolutional neural networks[J]. ICML, 2019.
> [2] Cihang Xie and Alan Yuille. Intriguing properties of adversarial training at scale. In ICLR, 2019.

---

### Official Review · AnonReviewer1 · 2020-10-28
**The motivating finding is interesting, but needs further investigation**

**Rating:** 4
**Confidence:** 4

**Review:**

Summary: This paper examines the robustness of adversarially robust models at the class-level.  Specifically, they note a disparity in the class-wise robustness of models for standard datasets. Furthermore, they suggest that many of these class-level vulnerabilities are eliminated if the model is trained without the corresponding confounding class. Finally, they propose a temperature based attack to further degrade accuracy of vulnerable classes.

Comments: I think that fine-grained studies of existing robust models can be valuable to identify ways to improve their performance.  While the paper is based on an interesting finding, the analysis is not sufficiently novel or extensive, and does not lead to a better understanding of why this phenomenon occurs or actionable insights to improve the performance of robust models.

Novelty: This paper is not the first to note the disparity in class-wise robustness---for instance, a fairly extensive study has been performed in https://arxiv.org/abs/2006.12621. Even if one were to consider both works as concurrent, the paper under review lacks depth and it is not clear what, if any, the impact of its findings are (see below). Relatedly, there is a line of work on adapting adversarial training to account for instance-level disparities in robustness---for instance, https://arxiv.org/abs/1910.08051.

Takeaways: The authors do not provide any conclusive evidence for the cause of class-wise disparities in robustness. They claim that this is linked to the norm of the weights, however they do not establish a causal link between the weights and per-class robustness (Fig. 4 could simply be correlation). Moreover, I expect that a similar relationship would be observed between standard and robust per-class accuracies (from Appendix Figure 6)---instead of the weight norm and robust accuracy---if both these quantities were normalized. In general, it is also tricky to make inferences based on the norm of the weights (without proper normalization) as the outputs of individual neurons in the pre-final layer could have vastly different scales. Additionally, the authors do not demonstrate that their findings can help to improve model robustness.  At the end of every section, the authors do provide some suggestions, however these are vague and unsubstantiated.

Overall, I think in its current form, this paper lacks depth and novelty. The observation of class-wise disparities alone is not substantial enough a contribution. As I mentioned above, the authors do not supplement this finding with a clear explanation of the underlying cause, or any evidence that this finding could help improve model robustness.

Additional comments:

* Are the confusion matrices based on targeted or untargeted attacks? If it is the later, then these numbers do not provide a complete picture of class-wise robustness. For an example of class i, it is possible that the adv. example that maximizes loss within an eps ball is from class j, though adversarial examples for many other examples exist within the ball (and are even closer to the original sample). The authors should repeat the evaluation using pairwise-targeted attacks.

* Many of the experimental choices are not sufficiently justified. For instance, why FGSM is used for ImageNet training, and why the best model is not picked based on cross validation. Further, the authors do not once mention what kind of adversary they consider.

* It is clear from Fig. 6 that, for most datasets, the same classes with lower robust accuracy have lower standard accuracy. Are the confusion matrices in Fig 2 based on all test set examples or just the ones that are correctly classified in the non-adversarial setting? What do the standard confusion matrices for these datasets look like?

* It is odd that the second element in the first row in Figure 3b (18) is larger than the corresponding element in Fig 3a. From what I understand, 3a should be an upper bound on every element in 3b.

* When evaluating the temperature based attack, do the authors apply the final perturbation to the original model (without the temperature scaling) or to the scaled model? If the numbers reported in Table 3 are based on the latter, then they do not indicate the robustness of the original model as the inference process has been modified.

#### Post-rebuttal Update ####

I thank the authors for their detailed response and edits to the paper. However, even after reading the rebuttal some of my original concerns stand:

[Novelty] As I mentioned in my original review, the discovery of disparities in class-wise robustness is not new to this paper. In the rebuttal the authors mention that their finding is different from [1] because in [1] the measure of class-wise robustness is distance to the decision boundary with respect to every other class. However, this is in my view, is just an alternative and well-established measure of robustness---i.e., distance to the decision boundary and robust accuracy are fairly correlated and not fundamentally different.

[Takeaways]
- The authors do not perform sufficient quantitative analysis to justify the link between robustness and weight norm. Moreover, without establishing the causality of this link, it is unclear to me how this observation provides any new insight to understand robust models.
- Figure 7 (in the revised manuscript) shows that a similar class-wise disparity is present even in the *standard accuracy* of *standard models*. Thus, although I find the observation of disparities in class-wise robustness interesting, I believe further investigation is needed to understand whether this is just an inherent property of the data distribution that hurts both standard and robust models or is specifically tied to robustness.
- The authors' comment about improved robustness based on methods adapted from [3] is misleading. As mentioned in Appendix G (I believe there is no Appendix H), this seems to be the case only for a specific attack. The authors themselves demonstrate a different attack under which the improved robustness of vulnerable classes disappears.

[Other comments] There is no evidence to suggest that untargeted attacks will find the *closest* adversarial example within an eps ball. The optimization problem for untargeted attacks is set up to maximize the loss (w.r.t. the ground truth label) and not to find the nearest misclassification. Thus, I still assert that to get a better picture of per-class robustness, the authors need to measure the targeted confusion matrix (or distance to per-class decision boundaries as in [1]).

Due to these concerns, I am unable to raise my score.

---

> ### Author Response · Authors · 2020-11-24
> **Response to AnonReviewer 1 (1/2): Answering Specific Questions**
>
> **Q(1) Cons about novelty.**
> **A(1)** Thanks for your comments. [1] and [2] are related to our work, however, [1] focuses on the distance from each example to the decision boundary, and [2] sets different iterative hyperparameters according to different examples, while our paper focuses on systematically studying the class-wise performance of robust model, e.g, the robust relationship between classes, the strength of attack in class-wise performance. Thus we believe our paper takes the first step to systematically investigate the role of different classes in adversarial training.
>
> **Q(2.1) Cons about takeaways (Lack of causality).**
> **A(2.1)** We believe that the discovery of the relation between class-wise robustness and the norm of classifier weight is meaningful, which can provide new insight to understand robust models. We also agree that it is very important to discover the causal link between this relation, and we will explore further in future work. We have modified the description of this part (causal link => related link).
>
> **Q(2.2) Cons about takeaways (similar norm relationship between standard and robust per-class accuracies).**
> **A(2.2)** The reviewer believes that in the robust model, standard accuracy (the follow-up is called accuracy) and robustness accuracy (the follow-up is called robustness) should have a similar relationship with the norm of classifier weight. This is true since one model has the same decision boundaries. However, we emphasize that if a model is a standard training model (not robust), then in these datasets, its accuracy and the norm of classifier weight are not positively correlated, because these datasets are sufficient for the standard model to be well trained.
>
> **Q(2.3) Cons about takeaways (the norm of classifier weight of different model have vastly different scales).**
> **A(2.3)** We normalize the norm of classifier weight and check that the norm of different models is in similar scales.
>
> **Q(2.4) Cons about takeaways (no actual robustness improvement).**
> **A(2.4)** We report a preliminary attempt based on [3] in Appendix H and results are shown in Table 5, which greatly improved robustness of vulnerable classes (under vanilla PGD attack).
>
> **Q(3) Cons about confusion matrices (targeted or untargeted attacks).**
> **A(3)** Confusion matrices are based on untargeted attacks. This is because untargeted attacks usually automatically find the nearest classification boundary. For example, the decision boundary of samples M (ground truth class: i) in $\varepsilon$-ball includes class i-class j (nearest) boundary and class i-class k (sub-nearest) boundary. When class j is removed, untargeted attacks will automatically attack the sample M to the current nearest decision boundary(class i-class k). However, Figure 3 shows that for many samples, class k does not exist. In other words, for many samples, the decision boundary in $\varepsilon$-ball is unique(only exist class i-class j boundary). In this case, even if the targeted attack is used, there will be no obvious improvement of attack rate, and we can prove this by comparing the results of MT[4] and vanilla PGD attack (i.e., from the perspective of absolute attack rate, MT is slightly better than vanilla PGD attack).
>
> [1] Nanda V, Dooley S, Singla S, et al. Fairness Through Robustness: Investigating Robustness Disparity in Deep Learning[J]. arXiv preprint arXiv:2006.12621, 2020.
> [2] Balaji Y, Goldstein T, Hoffman J. Instance adaptive adversarial training: Improved accuracy tradeoffs in neural nets[J]. arXiv preprint arXiv:1910.08051, 2019.
> [3] Tianyu Pang, Xiao Yang, Yinpeng Dong, Kun Xu, Hang Su, and Jun Zhu. Boosting adversarial training with hypersphere embedding. NeurIPS, 2020.
> [4] Gowal S, Uesato J, Qin C, et al. An alternative surrogate loss for pgd-based adversarial testing[J]. arXiv preprint arXiv:1910.09338, 2019.

---

> > ### Author Response · Authors · 2020-11-24
> > **Response to AnonReviewer 1 (2/2): Answering Specific Questions**
> >
> > **Q(4) Cons about experimental setting.**
> > **A(4)** As far as we know, [5] is one of the most popular methods for training models on ImageNet, among which FGSM is an important trick in their method. The epoch selection of all robustness models follows the original paper of each method. eg, it is known that overfitting is very serious in adversarial training, and 75-epoch on CIFAR-10 is used to obtain the highest robustness [6]. The choice of attackers also follows the original paper, which is described in detail in Section 3. For example, in CIFAR-10 & CIFAR-100 setup, ‘The training and testing attackers are ${\rm PGD}^{10}$/${\rm PGD}^{20}$.
> >
> > **Q(5) Cons about similar class-wise accurary/robustness between standard and robust per-class accuracies.**
> > **A(5)** At the end of Page 3, we emphasize that all experiments in the paper are based on robustness. We believe that standard training should mainly focus on accuracy (robustness is 0), and adversarial training should mainly focus on robustness. And Fig.6 emphasizes that the performance discrepancies among classes in the robust model (for robustness) are larger than that of the standard model (for accuracy), and we believe this phenomenon is worthy of attention.
> >
> > **Q(6) Cons about Figure 3.**
> > **A(6)** I am grateful to reviewer for careful observation. This is caused by a bug in the original code, and we have corrected and rechecked. The updated value still supports our conclusion.
> >
> > **Q(7) Cons about Temperature-PGD attack.**
> > **A(7)** The Temperature-PGD attack directly tests the models released by the original authors (the temperature is not scaled during training). We re-introduce the motivation of Temperature-PGD in Section 4. See updated paper for details.
> >
> > [5] Wong E, Rice L, Kolter J Z. Fast is better than free: Revisiting adversarial training[J]. arXiv preprint arXiv:2001.03994, 2020.
> > [6] Rice L, Wong E, Kolter J Z. Overfitting in adversarially robust deep learning[J]. arXiv preprint arXiv:2002.11569, 2020.

---

### Official Review · AnonReviewer2 · 2020-11-05
**detailed & interesting analysis**

**Rating:** 6
**Confidence:** 5

**Review:**

This paper presents a detailed analysis of adversarial training on several datasets (including CIFAR-10/100 and ImageNet), and observes three interesting (class-wise) properties: (1) some classes are extremely vulnerable to adversarial attacks; (2) for a certain class, its robustness is positively correlated to the norm of its FC layer's weight; (3) stronger attacks usually hurt vulnerable classes more. These results could be helpful for future works on furthering model robustness.


Pros:

(1) This paper is very clearly written and easy to follow.

(2) To the reviewer's best knowledge, this is the first attempt to analyze how adversarially trained models behave at the class level (rather than just reporting the overall performance on the whole dataset). And the observations are kind of surprising, e.g., there indeed exists certain classes that are much more vulnerable than others, and the observed confusion patterns (e.g., "class 9" and "class 1" in CIFAR-10 are easily confused with each other) are pretty consistent among different defense methods.


Cons:

The reviewer believes some conclusions of this paper can be further enhanced if analysis can be provided for the following parts:

(1) In Section 3.1, this paper finds certain (pairs of) classes are extremely vulnerable to adversarial attacks. As there are already some works that begin to take care of vulnerable samples (a more fine-grained level than vulnerable class) in adversarial training [1,2], it will be good to analyze if such customized strategies successfully reduce class-wise confusions under adversarial attacks.

(2) In Section 3.2, this paper observes that the robustness is correlated to the norm of weight, and hypothesis such phenomenon is caused by insufficient data in adversarial training. As there are already several works [3,4,5] on augmenting extra data for improving adversarial training, the authors can use these models to validate the hypothesis above. i.e., with more training data, will the robustness and the norm of weight become less related?

Besides, the reviewer is a little bit confused about the proposed attack in Section 4.  What is the motivation for adding the temperature parameter in PGD attack? Why it can hurt certain classes more? It seems that only empirical results are provided, but no motivation or explanations of why it can work are provided.  (which makes this section less related to the core parts of this paper and easily confuse readers)

The authors should carefully address these concerns during the rebuttal.


[1] Cheng M, Lei Q, Chen P Y, et al. Cat: Customized adversarial training for improved robustness[J]. arXiv preprint arXiv:2002.06789, 2020.

[2] Balaji Y, Goldstein T, Hoffman J. Instance adaptive adversarial training: Improved accuracy tradeoffs in neural nets[J]. arXiv preprint arXiv:1910.08051, 2019.

[3] Zhai R, Cai T, He D, et al. Adversarially robust generalization just requires more unlabeled data[J]. arXiv preprint arXiv:1906.00555, 2019.

[4] Carmon Y, Raghunathan A, Schmidt L, et al. Unlabeled data improves adversarial robustness[C]//Advances in Neural Information Processing Systems. 2019: 11192-11203.

[5] Uesato J, Alayrac J B, Huang P S, et al. Are labels required for improving adversarial robustness?[J]. arXiv preprint arXiv:1905.13725, 2019.

---

> ### Author Response · Authors · 2020-11-24
> **Response to AnonReviewer 2: Answering Specific Questions**
>
> **Q(1) Cons about Section 3.1.**
> **A(1)** Thanks for your constructive suggestions. We will conduct extended experiments and report the class-wise robustness of [1] and [2] in the future. In addition, we notice that [1] uses label smoothing technique, which makes this method may not be able to defense Temperature-PGD attack, see Section 4 in the updated paper.
>
> **Q(2) Cons about Section 3.2.**
> **A(2)** Although [3,4,5] use extra data to improve the model robustness, these extra data are not labeled. It is difficult for us to process the data differently by class. In addition, we check the relationship between class-wise robustness and the norm of weight in the RST model and find that it behaves similarly to Madry’s model (Figure 4(b)). The result is consistent with Table 2, we can find the class-wise robustness improvement of RST(PGD) relative to Madry’s model(PGD) is almost the same (i.e., each class has increased by about 10%), indicating that even with these extra data, the improvement of each class (limited by data) does not reach the upper bound. However, we notice that the improvement of HE-CMP in the vulnerable classes in Table 5 is significantly higher than that of the robust classes. We conduct experiments and find that the norm of classifier weight of vulnerable classes (i.e. class 3 and class 4) has significantly improved.
>
> **Madry’s model**
> **Class-wise robustness(Absolute value %):**
>
> | Class 0 | Class 1 | Class 2 | Class 3 | Class 4 | Class 5 | Class 6 | Class 7 | Class 8 | Class 9 |
> |  ----  | ----  |   ----  | ----  | ----  | ----  | ----  | ----  | ----  | ----  |
> | 57.6 | 74.9 |  37.6 | **22.1** | **30.7** | 41.2 | 55.6 | 64.0 | 71.0 | 62.0 |
>
> **Norm of classifier weight(Absolute value):**
>
> |Class 0| Class 1| Class 2| **Class 3**| **Class 4**| Class 5| Class 6| Class 7| Class 8| Class 9  |
> |  ----  | ----  |   ----  | ----  | ----  | ----  | ----  | ----  | ----  | ----  |
> |3.9399| 4.0705| 3.5359|**3.0687**| **3.5028**| 3.2904| 3.5884| 3.8438| 4.0630| 3.8535  |
>
> **Norm of classifier weight(Normalized value):**
>
> | Class 0| Class 1| Class 2| **Class 3**| **Class 4**| Class 5| Class 6| Class 7| Class 8| Class 9  |
> |  ----  | ----  |   ----  | ----  | ----  | ----  | ----  | ----  | ----  | ----  |
> | 0.96792| 1.0| 0.86867| **0.75388**| **0.86054**| 0.80835| 0.88157| 0.94429| 0.99814| 0.94669  |
>
> **HE-CMP**
> **Class-wise robustness(Absolute value %):**
>
> |Class 0| Class 1| Class 2|**Class 3**| **Class 4**| Class 5| Class 6| Class 7| Class 8| Class 9  |
> |  ----  | ----  |   ----  | ----  | ----  | ----  | ----  | ----  | ----  | ----  |
> |71.9| 78.9| 52.2| **51.6**| **71.8**| 59.3| 78.3| 70.7| 84. | 80.7  |
>
> **Norm of classifier weight(Absolute value):**
>
> |Class 0| Class 1| Class 2| **Class 3**| **Class 4**| Class 5| Class 6| Class 7| Class 8| Class 9|
> |  ----  | ----  |   ----  | ----  | ----  | ----  | ----  | ----  | ----  | ----  |
> |4.0331| 3.5757| 4.2669| **4.3090**| **4.2630**| 4.1211| 4.0952| 3.8379| 3.9430| 3.7932|
>
> **Norm of classifier weight(Normalized value):**
>
> |Class 0| Class 1| Class 2| **Class 3**| **Class 4**| Class 5| Class 6| Class 7| Class 8| Class 9|
> |  ----  | ----  |   ----  | ----  | ----  | ----  | ----  | ----  | ----  | ----  |
> |0.93598| 0.82981| 0.99022| **1.0**|**0.98932**| 0.95640| 0.95040| 0.89069| 0.91507| 0.88030|
>
> **Q(3) Cons about Section 4.**
> **A(3)** The output probability distribution of vulnerable classes is relatively smooth (i,e, vulnerable classes has a smaller variance, Figure 5(a) and 5(e)), which makes vanilla PGD attacks circle around multiple decision boundaries. Temperature-PGD can create virtual power to move the example to the nearest decision boundary. We re-introduce the motivation of Temperature-PGD in Section 4. See updated paper for details.
>
> [1] Cheng M, Lei Q, Chen P Y, et al. Cat: Customized adversarial training for improved robustness[J]. arXiv preprint arXiv:2002.06789, 2020.
> [2] Balaji Y, Goldstein T, Hoffman J. Instance adaptive adversarial training: Improved accuracy tradeoffs in neural nets[J]. arXiv preprint arXiv:1910.08051, 2019.
> [3] Zhai R, Cai T, He D, et al. Adversarially robust generalization just requires more unlabeled data[J]. arXiv preprint arXiv:1906.00555, 2019.
> [4] Carmon Y, Raghunathan A, Schmidt L, et al. Unlabeled data improves adversarial robustness[C]//Advances in Neural Information Processing Systems. 2019: 11192-11203.
> [5] Uesato J, Alayrac J B, Huang P S, et al. Are labels required for improving adversarial robustness?[J]. arXiv preprint arXiv:1905.13725, 2019.

---

### Author Response · Authors · 2020-11-24
**Summary of Changes in Revision**

We thank all reviewers for their valuable feedback.

We updated our submission with the following key changes:
* Adjust the relationship between class-wise robustness and the norm of classifier weight from causal link to related link. We will explore their causal link in future work. But we emphasize that this phenomenon itself also helps researchers better understand the robust model.
* Re-introduce the motivation and mechanism of Temperature-PGD, and we emphasize that this attack can help us better understand the defense mechanism of some state-of-the-art models from the class-wise perspective.
* Add some related work provided by reviewers.
* Update and recheck Figure 3(b).

---

### Decision · Program_Chairs · 2021-01-07
**Final Decision**

**Decision:**

Reject

**Comment:**

Analyzing class-wise adversarial vulnerability of models is an interesting direction to pursue. However, the authors should consult the references pointed out in the reviews to put their contributions in the right perspective. Overall, the lines of inquiry explored in this paper are of interest but, as some of the reviewers point out, there are improvement in the methodology that still need to be addressed before this paper is ready for publication. (I very much recommend that the authors do build on this feedback and revise the paper, as it will be a valuable contribution then.)